# Map of Enteropathogenic *Escherichia coli* Targets Mitochondria and Triggers DRP-1-Mediated Mitochondrial Fission and Cell Apoptosis in Bovine Mastitis

**DOI:** 10.3390/ijms23094907

**Published:** 2022-04-28

**Authors:** Yanan Li, Yaohong Zhu, Bingxin Chu, Ning Liu, Shiyan Chen, Jiufeng Wang, Yunjing Zou

**Affiliations:** Department of Clinical Veterinary Medicine, College of Veterinary Medicine, China Agricultural University, Beijing 100193, China; 13796685756@163.com (Y.L.); zhu_yaohong@hotmail.com (Y.Z.); barrylao@163.com (B.C.); nliu2224@163.com (N.L.); chen_shiyan1993@163.com (S.C.)

**Keywords:** EPEC, DRP-1, *map*, apoptosis, bovine mastitis

## Abstract

Bovine mastitis seriously affects bovine health and dairy product quality. *Escherichia coli* is the most important pathogen in the environment and dairy products. Enteropathogenic *Escherichia coli* (EPEC) is a zoonotic pathogen, which seriously threatens the health of people and dairy cows. We recently reported that *E. coli* can induce endogenous apoptosis in bovine mammary epithelial cells. However, the mechanism of EPEC-damaged mitochondria and -induced bovine mastitis is unclear. In this study, we found that EPEC can induce DRP-1-dependent mitochondrial fission and apoptosis. This was verified by the application of Mdivi, a DRP-1 inhibitor. Meanwhile, in order to verify the role of the Map virulence factor in EPEC-induced bovine mastitis, we constructed a *map* mutant, complementary strain, and recombinant plasmid Map_His_. In the present study, we find that Map induced DRP-1-mediated mitochondrial fission, resulting in mitochondrial dysfunction and apoptosis. These inferences were further verified in vivo by establishing a mouse mastitis model. After the *map* gene was knocked out, breast inflammation and apoptosis in mice were significantly alleviated. All results show that EPEC targets mitochondria by secreting the Map virulence factor to induce DRP-1-mediated mitochondrial fission, mitochondrial dysfunction, and endogenous apoptosis in bovine mastitis.

## 1. Introduction

Bovine mastitis, a substantial inflammation of the mammary gland, seriously affects the health of dairy cows and the production of dairy products worldwide, and even threatens the public health system [1]. However, in recent decades, due to the importance attached to the dairy farming industry, the prevalence of contagious mastitis has significantly decreased, and the incidence of environmental mastitis has increased relatively or absolutely [2]. One of the most important environmental pathogens is *Escherichia coli* (*E. coli*), which usually attacks the mammary gland in the early stages of lactation. If it is not treated in time, it can even cause fatal consequences [3]. As early as 1994, it was reported that feces were the main source of environmental pathogens [4]. The investigation on the prevalence of *E. coli* in cattle farms showed that enteropathogenic *Escherichia coli* (EPEC) was detected in the feces of healthy cattle and diarrhea cattle [5,6]. EPEC is also considered a zoonotic pathogen, which can cause persistent diarrhea. It is related to high morbidity and mortality in developing countries and sporadic and severe outbreaks in developed countries. Studies have shown that *E. coli* can be detected in raw milk and some cheese products, which is a threat to human health [7]. Thus, it is quite necessary for the pathogenic mechanism research and subsequent prevention and control measures of EPEC.

EPEC is widely regarded as a stealth pathogen, which relies on type-III secretory systems (T3SS) to directly inject a dozen effectors into host cells to escape a mucosal immune response, thereby persisting and spreading between hosts [8]. These effectors can infect cells by hijacking the cytoskeleton and different organelles, and interfering with the process of cell signal transmission [9]. To date, a total of seven effector proteins encoded by the LEE virulence island have been found, namely Tir, EspB, EspF, EspG, EspH, EspZ (formerly SepZ), and Map (the description of these virulence factors is detailed in Appendix A). All LEE effector proteins, except EspZ, have been proved to have harmful effects on host cells, among which Tir, Map, and EspF are studied in-depth [10,11]. Map and EspF have many functions, and their functions partially overlap, such as targeting mitochondria and destroying the tight junction between epithelial cells [12,13]. EspF targets mitochondria to initiate host cell apoptosis [14] and inhibit phagocytosis [15], whereas Map can induce the formation of filamentous feet in a way independent of mitochondrial targeting [16]. In addition, EspF has been shown to not only target mitochondria, but also nucleoli, resulting in the complete loss of nucleolin [14]. Each effector protein may target different organelles to change different cell processes. Therefore, in the process of EPEC-induced bovine mastitis, the determination of which effector proteins play a pathogenic role should be investigated.

Mitochondria are the center of energy production, which can maintain cell homeostasis by regulating a series of cellular processes. According to the situation of cells, mitochondria undergo different degrees of fusion and fission, which makes mitochondria highly dynamic. The fusion is divided into two parts. The outer mitochondrial membrane (OMM) fusion is mediated by mitofusin-1/2 (MFN-1/2) homodimer/heterodimer and the inner mitochondrial membrane fusion is mediated by optic atrophy 1 (OPA1) homodimer [17], whereas the fission is mediated by the recruitment of dynamin-related protein 1 (DRP-1) to the OMM [18]. Under normal circumstances, the two opposite processes are to maintain dynamic balance; however, once in a pathological condition, the balance is broken, which tends to fission and eventually leads to apoptosis [19]. In the view of the significance of mitochondria in regulating cell function, many pathogens have evolved mechanisms to hijack the mitochondrial apoptosis pathway; that is, some can prevent apoptosis to maintain their own colonization, and some can promote apoptosis to escape immune clearance or increase the area of infection [20]. Studies have shown that DRP-1 can induce apoptosis by stimulating BAX oligomerization and Cytochrome C (Cyt-C) release [21]. In addition, hepatitis C virus can promote sustained infection through DRP-1-mediated mitochondrial fission [22]. Then, in consideration of EPEC-induced bovine mastitis, a further investigation is required to determine if EPEC can induce the apoptosis of breast epithelial cells through effector-protein Map to maintain the continuous infection, as well as if DRP-1-mediated excessive mitochondrial fission is involved in the process of apoptosis.

We hypothesize that EPEC leads to mitochondrial pathway apoptosis by secreting effector-protein Map in bovine mastitis. Using the method of cell biology, this study aims to explore the mechanism of damage to host cells caused by effector-protein Map of EPEC in the process of bovine mastitis. Simultaneously, a mouse mastitis model was established to verify the effect of effector-protein Map in vivo. It is worth noting that our data provide a theoretical basis for better understanding the zoonotic pathogen-EPEC pathogenic mechanism and the prevention of mastitis.

## 2. Results

### 2.1. E. Coli (EPEC) Induces Increased Mitochondrial Fission, Decreased Fusion, Cyt-C Release, and Decreased MMP, Eventually Leading to Apoptosis

In previous studies, our research group found that *E. coli* can induce apoptosis. Once the function of mitochondria as energy centers is damaged, it induces a series of problems, including apoptosis. Since mitochondria are highly dynamic, we first detected the changes of key proteins related to mitochondrial dynamics, such as DRP-1 and MFN-1/MFN2. We found that *E. coli* (EPEC) increased the DRP-1 expression and decreased the MFN-1/MFN2 expression (Figure 1A). Due to the dephosphorylation at the ser637 site, a key to mitochondrial fission, we measured the p-DRP-1 (ser637) level. The results show that *E. coli* (EPEC) could reduce the level of p-DRP-1 (ser637) (Figure 1A), indicating that it promotes the dephosphorylation of DRP-1 at the ser637 site. It is reported that if the dynamic balance of mitochondria is broken, apoptosis can be induced [23]. We detected the protein expression of caspase-3, the gold index of apoptosis. The Western blot result demonstrated that *E. coli* (EPEC) increased the level of caspase-3 p17 (Figure 1A), indicating the occurrence of apoptosis. Given the mitochondrial fission induced by *E. coli*, it can be determined that it caused mitochondria Cyt -C release and the decrease in mitochondrial membrane potential (MMP). We successfully isolated the mitochondria and used them to detect the content of Cyt-C. It was found that *E. coli* (EPEC) caused mitochondrial Cyt-C to be released into the cytoplasm (Figure 1B). Rhodamine 123 staining showed that the green fluorescence decreased or even disappeared, meaning that *E. coli* (EPEC) caused the decrease in MMP (Figure 1C).

### 2.2. E. Coli (EPEC) Mediates DRP-1-Dependent Apoptosis

DRP-1 is mainly located in the cytoplasm. If it is transferred from the cytoplasm to the OMM, it leads to mitochondrial fission [24]. We applied Mdivi (a selective DRP-1 inhibitor) to verify the critical role of DRP-1 in *E. coli* (EPEC)-induced apoptosis. Immunofluorescence images showed that, compared to the CONT group, yellow fluorescent spots in the ECOL group significantly increased, and the addition of Mdivi reduced the formation of yellow spots (Figure 2B). Subsequently, the changes in the MMP were detected by JC-1. The results in Figure 2C show that the red fluorescence is significantly weakened and the green fluorescence is enhanced in the ECOL group. In comparison to the ECOL group, the red fluorescence in the ECOL + Mdivi group was significantly enhanced and the green fluorescence was significantly weakened (Figure 2C), indicating that *E. coli* (EPEC) can reduce the MMP, and the addition of Mdivi can lessen this decrease. Finally, we detected the occurrence of apoptosis. The Western blot results suggest that the expression of caspase-3 p17 significantly increases in the ECOL group (Figure 2A). In comparison to the ECOL group, the caspase-3 p17 level in the ECOL + Mdivi group significantly decreased. The immunofluorescence results are consistent with the Western blot results. The red spots significantly increase in ECOL group (Figure 2D). After adding Mdivi, the red spots significantly decrease (Figure 2D). The nucleus was stained by DAPI. The nucleus collapsed and was deeply stained in the ECOL group, while the state of the nucleus recovered after adding Mdivi (Figure 2D). These results imply that *E. coli* (EPEC) can induce apoptosis, and Mdivi can alleviate *E. coli*-induced apoptosis.

### 2.3. E. Coli (EPEC) Evokes the Continuous Opening of mPTP and Mitochondrial Cyt-C and Ca^2+^ Release

It has been reported that continuous mitochondrial permeability transition pore (mPTP) opening can induce apoptosis [25]. When mPTP continues to open, Cyt-C and Ca^2+^ in mitochondria are released into the cytoplasm. As shown in Figure 3B, the decrease in mitochondrial calcein fluorescence during *E. coli* (EPEC) exposure represents the opening of mPTP. When CsA was administered an mPTP inhibitor, the increase in mitochondrial calcein fluorescence represented that mPTP was turned off. The mitochondria were isolated to detect Cyt-C levels in the mitochondria and cytoplasm. In the ECOL group, the content of Cyt-C in the mitochondria significantly decreased, while the content of Cyt-C in the cytoplasm significantly increased (Figure 3C). After the introduction of CsA, the Cyt-C level in the cytoplasm significantly decreased and the Cyt-C level in the mitochondria significantly increased in the ECOL + CsA group (Figure 3C). This indicates that, when mPTP is turned off, *E. coli* (EPEC)-induced Cyt-C release can be attenuated. The immunofluorescence images showed that almost all Cyt-C (green) and mitochondria (red) were co-located in normal cells and appeared yellow (Figure 3D). Yellow spots basically disappeared in the ECOL group, whereas they significantly increased in the ECOL + CsA group (Figure 3D). These results signify that *E. coli* (EPEC) induces the release of Cyt-C, and CsA can reverse this phenomenon. The increase in Ca^2+^ concentration is also one of the signals of mPTP opening. Flow cytometry analysis showed that *E. coli* (EPEC) could lead to the increase in Ca^2+^ concentration, and CsA could alleviate the trend (Figure 3E). The result of the cleaved caspase-3 level (Figure 3A) indicates that *E. coli* (EPEC)-induced apoptosis can be reversed by CsA.

### 2.4. E. Coli (EPEC)-Map Induces Mitochondrial Fragmentation

To identify whether the effector-protein Map secreted by *E. coli* (EPEC) is the main reason for inducing mitochondrial fission, a *map* mutant (∆*map*) and *map* gene complementation strain (∆*map* + *pmap*) were constructed. In comparison to the WT strain, the ∆*map* strain induced less DRP-1 protein expression and more MFN-1/MFN-2 and p-DRP-1 ser637 protein expression (Figure 4A). The further detection of DRP-1 translocation showed that the ∆*map* strain alleviated DRP-1 translocation to mitochondria (Figure 4B). In addition, mitochondrial staining (TOM20, green) was used to observe the mitochondria morphology. In normal cells, mitochondria present a network structure (Figure 4C). The increase in network and individual counts, and the decrease in the mean rod/branch, the mean branch per network length, and the mitochondrial footprint (Figure 4D) all represent that the WT strain can induce a reduction in the mitochondrial network structure and replace it with more point mitochondria. When the effector-protein Map was knocked out, some mitochondrial network structures recovered (Figure 3C,D). When the effector-protein Map was complemented in the ∆*map* mutant, all these changes caused by the ∆*map* strain were reversed.

### 2.5. E. Coli (EPEC)-Map Results in Sustained mPTP Opening, Cyt-C Release, Low MMP, and Apoptosis

Mitochondrial fission and fragmentation can induce mitochondrial dysfunction and apoptosis. In view of this, we conducted experiments for the MMP, mPTP, and apoptosis. JC-1 staining analysis showed that red fluorescence increased and green fluorescence decreased after the knockout of the *map* gene (Figure 5B). In comparison to the WT strain, effector-protein Map may be the main reason for the decrease in the MMP caused by *E. coli* (EPEC). The results of the mPTP kit and Cyt-C staining show that the ∆*map* strain causes more mPTP closure and Cyt-C mitochondrial co-localization (Figure 5D,F). We further detected apoptosis and found that the ∆*map* strain resulted in a lower expression of pro-apoptotic proteins BAX and cleaved caspase-3, and a higher expression of anti-apoptotic protein BCL-2 (Figure 5A). Immunofluorescence and confocal microscopy allowed us to observe that the ∆*map* strain caused less cleaved caspase-3 accumulation than the WT strain (Figure 5C). The same results were obtained by using flow cytometry to analyze the apoptosis rate. The apoptosis rate of the ∆*map* strain was significantly reduced, compared to the WT strain (Figure 5E). All these results indicate that the *map* gene is an important virulence factor for *E. coli* (EPEC) to induce apoptosis. The changes brought about by *map* gene knockout were destroyed by the *map* gene complementation strain. Once again, it shows the importance of the *map* gene in *E. coli* (EPEC)-induced apoptosis.

### 2.6. Map, as a Virulence Factor of E. Coli (EPEC), Is Sufficient to Induce Mitochondrial Fission and Apoptosis

To visualize the intracellular localization of Map during infection, we synthesized a His-flag-tag on *map* in the pc DNA3.1 plasmid. First, we performed staining for the His-flag-tag and mitochondria. Interestingly, the *map* (His) and mitochondria co-localization was observed by a confocal microscope (Figure 6A), and mitochondria also showed a trend of fragmentation. Then, we observed the translocation of DRP-1 and found that *map* was sufficient to induce the DRP-1 translocation to mitochondria and mitochondrial fission (Figure 6F). This is consistent with the results of protein detection. *Map* induced more DRP-1 and less MFN-1/MFN-2 and p-DRP-1 ser637 levels (Figure 6B). JC-1 staining and mPTP kits were used to evaluate the mitochondrial function. The results show that effector-protein Map can lead to the reduction in MMP and the opening of mPTP (Figure 6D,E), indicating the induction of mitochondrial dysfunction. The apoptosis rate was further detected by flow cytometry analysis. The results show that effector-protein Map causes a significant increase in the apoptosis rate (Figure 6G), which is consistent with the results of Western blot (increased cleaved caspase-3 and BAX, decreased BCL-2, Figure 6C). These results indicate that effector-protein Map is sufficient to induce apoptosis.

### 2.7. Map Is Required for E. Coli (EPEC)-Induced Apoptosis In Vivo

To verify whether the effector-protein Map can also play the same cytotoxicity in vivo, we established a mouse mastitis model. H & E staining was performed on breast tissue to evaluate breast histopathological damage. It was found that the WT strain could significantly affect histopathological damage (thickened mammary acinus wall and filled with inflammatory cells in acinus), and the ∆*map* strain could reverse a certain amount of these changes (Figure 7E,F). The detection of protein levels showed that the WT strain caused the increase in mitochondrial fission and the decrease in fusion in mouse breast epithelial cells (Figure 7A). Furthermore, the cell ultrastructure was observed by TEM. It was discovered that the WT strain induced mitochondria to swell, dissolve, or even disappear; disordered the arrangement of the endoplasmic reticulum (ER); and fragmented the nucleus (Figure 7D). In the ∆*map* group, the state of the mitochondria, ER, and nucleus recovered, but this positive change was destroyed by the ∆*map* + *pmap* strain (Figure 7D). Finally, the apoptosis was evaluated. TUNEL staining presents that, in comparison to the WT group, the proportion of TUNEL-positive cells in the ∆*map* group significantly decreased; however, when the *map* gene was supplemented, the positive proportion significantly increased (Figure 7C). Consistent with the TUNEL staining results, the ∆*map* strain induced less cleaved caspase-3 and BAX protein expression and more BCL-2 protein expression, compared to the WT/∆*map* + *pmap* strain (Figure 7B).

## 3. Discussion and Conclusions

A growing number of viral and bacterial proteins are considered to target host mitochondria and play an important role in the pathogenesis as regulators of apoptosis [22]. Map, as an important virulence factor of EPEC, has been reported to induce host cell apoptosis, but its specific pathogenic molecular mechanism is not clear in bovine mastitis. In this study, we proposed for the first time that the *map* gene can induce DRP-1-mediated mitochondrial fission, leading to mitochondrial dysfunction and apoptosis in EPEC-induced bovine mastitis.

### 3.1. EPEC Leads to DRP-1-Dependent Endogenous Apoptosis in Bovine Mastitis

Bovine mastitis has always been a persistent problem in the world, which leads to the premature elimination of bovine, the decline of milk quality, and serious economic losses. *E. coli* is a common pathogen of bovine mastitis, which can cause local or acute mastitis. However, the pathogenic *E. coli* in the environment is easy to be ignored. EPEC, as a common environmental *E. coli* pathogen, widely exists in water sources, feces, workers’ clothes, and milk samples with mastitis in dairy farms [26]. When bacterial infection leads to acute inflammation, the host generally phagocytizes pathogens through neutrophils to resist infection [27]. Furthermore, it secretes inflammatory factors, such as interleukin and TNF-α, in response to inflammatory stimulation, which also triggers apoptosis [28]. As a way of regulating cell death, apoptosis not only plays a crucial role in the dynamic balance of organisms, but is also a natural means of biological defense. Some bacteria, such as *Salmonella*, have evolved ways to escape to maintain their “niche” by hijacking the apoptosis pathway [29]. Some bacteria can also maintain persistent infection by promoting apoptosis. In this study, MAC-T cells infected with EPEC were characterized by nuclear pyknosis, deep staining, and even disintegration, indicating apoptotic death (Figure 2, Figure 3, Figure 4, Figure 5 and Figure 6). Apoptosis is triggered by endogenous (mitochondrial) or exogenous (cell-surface receptor) signal pathways, and finally induces morphological changes of apoptosis through caspase-3 activation. Here, it is argued that EPEC induces apoptosis through the mitochondrial pathway, since we showed that EPEC can induce the decrease in MMP and the continuous opening of mPTP (Figure 1, Figure 2 and Figure 3), and lead to the escape of Cyt-C and Ca^2+^ from the mitochondria to cytoplasm, thus activating caspase-3 (Figure 1 and Figure 3). These data demonstrate that EPEC promotes mitochondrial Cyt-C release, reduces MMP, and finally causes mitochondrial apoptosis. This is consistent with the previous studies that determined the EPEC can induce apoptosis [30]. Meanwhile, the mPTP opening can induce apoptosis. The introduction of CsA reversed EPEC-induced apoptosis (Figure 3). These data argue that EPEC-induced apoptosis is closely related to the continuous opening of mPTP. Above all, there is a considerable correlation between acute bacterial infection and apoptosis. The focus of this study is to explore the molecular mechanism of the mitochondrial apoptosis pathway in acute EPEC-infected bovine mastitis.

Mitochondria are the center of energy function and can determine the life and death of cells. Mitochondria undergo considerable morphological changes in the process of apoptosis, such as mitochondrial fission and ridge remodeling, which promotes the release of Cyt-C or other pro-apoptotic proteins, and eventually leads to caspase activation and cell death [31]. The maintenance of mitochondrial function lies in the balance of mitochondrial dynamics. Nevertheless, some pathogens have evolved mechanisms to hijack mitochondria to escape immunity. *Pseudomonas aeruginosa* uses its T3SS to secrete an effector ExoT, which can induce an increase in the expression of pro-apoptotic proteins BAX, Bid, and Bim and the decrease in MMP, thus activating the inherent mitochondria cell-death pathway [32]. *Helicobacter pylori* can also target mitochondria by secreting vacuolar cytotoxin VacA, thereby inducing mitochondrial fragmentation and Cyt-C release [33]. *Legionella pneumophila* secretes effector-protein MitF to induce DRP1-dependent mitochondrial fission in macrophages. If blocking mitochondrial fragmentation, *Legionella pneumophila* replication is reduced in lung cells [34]. All these reports indicate that mitochondrial dynamics plays a key role in apoptosis induced by bacterial infection. Consistent with these studies, our present study found that EPEC infection increased the expression of DRP-1 and decreased the expression of MFN-1/MFN-2 (Figure 1, Figure 2 and Figure 4), indicating that EPEC could cause excessive mitochondrial fission as well as reduce fusion. Similarly, the decrease in p-DRP-1 ser637 levels (Figure 1 and Figure 4) suggests the increase in DRP-1 mitochondrial translocation, which can also be reflected by the co-localization of DRP-1 and mitochondria (Figure 2). It suggested that EPEC could promote the dynamic changes of mitochondria tending to fission. Using Mdivi to inhibit mitochondrial fission, EPEC-induced apoptosis can be reversed and mitochondrial morphology can be partly restored (Figure 2). Interestingly, after the application of Mdivi, the low MMP induced by EPEC was slowed down (Figure 2), suggesting that the mitochondrial fission of EPEC-infected cells may be a trigger for mitochondrial dysfunction. Those phenomena indicate that EPEC infection induces apoptosis through mitochondrial fragmentation. However, as mentioned above, bacteria induce mitochondrial fission through secreted virulence, so the effectors that play a role in EPEC infection remains a mystery.

### 3.2. The Map Virulence Factor Induces Mitochondrial Fission and Mitochondrial Apoptosis in EPEC-Induced Bovine Mastitis

Although EPEC is an extracellular bacterium that cannot enter the cell for direct signal interference, it can secrete multiple effectors through its T3SS to achieve cytotoxicity. Based on our findings that EPEC can induce apoptosis in the mitochondrial pathway, we purposely screened effector proteins that are related to mitochondria or apoptosis. Studies have clarified that EspF has an N-terminal mitochondrial-targeting sequence, which can be located in the mitochondrial matrix. Therefore, it can induce MMP loss, caspase activation, and final apoptosis [35,36]. It has been pointed out that Map can also target mitochondria to induce Cyt-C release and apoptosis [12,37]. Cif can also induce delayed apoptosis by affecting the cell cycle [38]. However, unlike EspF and Cif, studies have shown that Map may also be related to mitochondrial morphological changes [39]. Moreover, most of the studies on EPEC are focused on the gastrointestinal tract. Due to the different host/cell, the effects of effectors are very different. Due to this, our study mainly focused on the role of effector Map in EPEC-induced apoptosis in bovine mastitis. We established a *map* mutant and complemented strains to explore the molecular mechanism of *map* as a virulence factor. In this study, it was observed that the effect of the *map* mutant on mitochondrial morphology was much weaker than that of the WT strain (Figure 4). Specifically, more fusion, less fission, and the mitochondrial translocation of DRP-1 in the *map* mutant were observed (Figure 4). It can also be observed from the quantitative analysis of mitochondrial morphology that the mitochondrial network structure recovered (Figure 4). This indicated that the knockout of the *map* gene could alleviate the excessive mitochondria fission and fragmentation caused by the WT strain. Consistent with the previous studies, the *map* mutant can slow down the decrease in MMP [39]. We also found that the *map* mutant could alleviate the sustained opening of mPTP and the mitochondrial release of Cyt-C, as well as subsequent apoptosis (Figure 5). Disturbingly, the *map* mutant complemented with *map* infection presented all phenotypes of the WT strain (Figure 4 and Figure 5), emphasizing that Map is indeed a key virulence factor for EPEC to trigger these phenotypes. Taken together, these effects on mitochondria and apoptosis once again emphasize the importance of EPEC Map, suggesting that EPEC uses Map as an effective virulence factor to induce the main disturbance of host cell homeostasis. Since EPEC is an extracellular bacterium and Map is necessary to induce host cytotoxicity, we intended to know whether effector-protein Map alone could be enough to induce cytotoxicity during infection. Meanwhile, we were also concerned about whether Map plays a role in targeting mitochondria. The co-localization of Map (His) and mitochondria showed that Map could indeed target mitochondria (Figure 6A). The effect of effector-protein Map alone is enough to induce DRP-1 translocation, mitochondrial dysfunction, and apoptosis (Figure 6). All these proved the virulence of Map again. Interestingly, even in the absence of *map*, mitochondrial dysfunction (MMP, mPTP), Cyt-C release, and apoptosis still occurred, albeit to a minor extent. This suggests that other effectors may also target these epithelial cell functions and play a synergistic role with *map*. For example, EspF may be a candidate because it can target mitochondria and trigger mitochondrial apoptosis [40]. Future studies will explore whether Map and EspF can overlap in the induction of apoptosis.

All of the above conclusions are base on our application of bovine mammary epithelial cells as an in vitro infection model to explore the role of Map. However, whether it performs the same effect in vivo has not been addressed. Thus, we established an EPEC-infected mouse mastitis model to explore the role of Map in vivo. Consistent with the results in vitro, Map can indeed induce mitochondrial excessive fission, reduced fusion, mitochondrial damage (mitochondrial swelling or even disappearance), and apoptosis (Figure 7). Meanwhile, the knockout of the *map* gene could also reduce the breast inflammation induced by WT EPEC. This is in agreement with the previous studies, which determine that Map can also cause pathological damage to epithelial cells [41]. These data mean that Map, as a virulence factor, can also cause mitochondrial dysfunction and apoptosis in vivo. These results are consistent with those in vitro, denoting that the Map virulence factor is required for EPEC to induce mitochondrial damage and apoptosis.

In summary, we proposed a model (graphical abstract) where Map, by targeting mitochondrial dynamics, induced mitochondrial dysfunction and provoked endogenous apoptosis. Therefore, our results strongly suggest that the regulation of mitochondrial dynamics induced by bacteria during infection forms the host cell death response and represents the key virulence strategy of EPEC-induced bovine mastitis.

## 4. Materials and Methods

### 4.1. Bacterial Strains

The *E. coli* wild-type (WT) strain (EPEC, O111:K58, CVCC 1450) is a commercial strain and was purchased from the China Veterinary Culture Collection Center. Our research group sequenced its whole genome and found that it has no drug-resistance gene. The *map* mutant (∆*map*) strain was a derivative of the parental *E. coli* WT strain constructed using the CRISPR-Cas9 system (a kind gift from Prof. Sheng Yang, Shanghai Institutes for Biological Sciences), as previously described [42]. The *map* gene complementation strain (∆*map* + *pmap*) was a derivative of the ∆*map* strain constructed with a pUC 19 plasmid (kind gift of Dr. Ning Xie, China Agricultural University) using 2 × Seamless Cloning Mix kit (CL117-01, Biomed, Beijing, China). The WT and ∆*map* strains were grown in Luria–Bertani (LB) broth, while the ∆*map* + *pmap* strain was grown in LB broth containing ampicillin.

### 4.2. Cell Culture

MAC-T cells were plated in the 6- or 24-well flat-bottom cell culture plates and cultured in DMEM (Life Sciences, St. Petersburg, FL, USA) supplemented with 10% FBS (Thermo Fisher Scientific, Waltham, MA, USA) and 1% penicillin/streptomycin in a CO_2_ incubator (37 °C and 5% CO_2_) for 24 h. MAC-T cells were washed with PBS (3 times) to remove antibiotics and exposed to WT, ∆*map*, and ∆*map* + *pmap* strains for 8 h (MOI of all strains = 66).

MAC-T cells performed the following operations to verify the role of DRP-1 and mPTP in *E. coli* (EPEC) infection. In brief, MAC-T cells were pre-treated with Mdivi (10 μM, HY-15886, a selective inhibitor of DRP-1 activity from MCE, Princeton, NJ, USA)/CsA (5μM, C8781, a selective inhibitor of mPTP, Solarbio, Beijing, China) for 4/2 h, washed with PBS (3 times) to remove inhibitors, and exposed to *E. coli* (EPEC, MOI = 66) for 8 h.

### 4.3. Western Blot

Cell and tissue total protein were obtained by an RIPA buffer (Sigma-Aldrich, St. Louis, MO, USA. Moreover, mitochondria and cytoplasms were extracted with a kit (C3601, Beyotime Biotechnology, Shanghai, China). Each group of 25 μg protein samples was transferred to the PVDF membrane by SDS-PAGE. Then, the membrane was blocked with 5% skimmed milk for 1 h (37 °C). Next, antibody incubation was performed. BAX, BCL-2, caspases-3/p17/p19, VDAC1/2, MFN-1, and MFN-2 antibodies were used at a dilution of 1:1000, and GAPDH, β-actin, and Tubulin antibodies were used in dilution of 1:10,000 (all these were purchased from Proteintech Group, Rosemont, IL, USA). DRP-1 and Cyt-C antibodies were purchased from Abcam (Cambridge, UK) and were used at dilution 1:1000. Phospho-DRP-1 S637 (1:1000, AP0812) antibody was purchased from ABclone (Wuhan, China). Cleaved caspase-3 antibody was purchased from Cell Signaling Technology (Beverly, MA, USA) and was used at dilution 1:1000. Finally, the corresponding secondary antibody was incubated. Protein-band visualization and analysis were used by the ECL detection system (Tanon, Shanghai, China) and ImageJ software (version 1.50).

### 4.4. Immunofluorescence

First, the mitochondria were labeled with Mito-Tracker Red CMXRos for 30 min. (37 °C, C1035, Beyotime Biotechnology, Shanghai, China). Second, the MAC-T cell climbing sheets were fixed with 4% paraformaldehyde (10 min), and then permeabilized with 1% Triton-X-100 (13 min). Third, they were blocked with 2% BSA at room temperature (1.5 h). Fourth, they were incubated with anti-DRP-1 (1:200, Abcam, Cambridge, UK), anti-Cyt-C (1:100, Abcam, UK), and anti-cleaved Caspase-3 (1:500, Cell Signaling Technology, Danvers, USA). Fifth, cells were stained with corresponding secondary antibody for 1 h. Sixth, DAPI (C0060, Solarbio, Beijing, China) was used to stain the nucleus for 7 min. Finally, images were captured and observed by the Nikon A1 confocal laser scanning microscope.

### 4.5. Mitochondrial Imaging

Mitochondria were stained by anti-TOM20 (1:300, Proteintech Group Inc., USA) according to the above immunofluorescence method. Mitochondrial images were obtained by a confocal laser scanning microscope (Nikon A1, Tokyo, Japan), and the mitochondrial network morphology was analyzed by FIJI (version 2.0.1) with Mina plug-in, as previous mentioned [43].

### 4.6. MMP and mPTP Measurements

MMP was detected with JC-1 (C2003S) and Rhodamine 123 (C2008S) kits purchased from Beyotime Biotechnology (China). JC-1 monomers (green fluorescence) and JC-1 aggregates (red fluorescence) represent a low MMP and high MMP, respectively. Rhodamine 123 shows a green fluorescence, which represents a high MMP, whereas the decrease or disappearance of green fluorescence represents a low MMP.

The opening degree of mPTP was detected by an mPTP kit (C2009S, Beyotime Biotechnology, China). In short, calcein shows a green fluorescence when entering the cell. Once exposed to CO^2+^, the green fluorescence is quenched. When mPTP is close, calcein is loaded onto the mitochondria and does not make contact with CO^2+^, resulting in the presentation of green fluorescence. When mPTP is open, calcein flows out of the mitochondria into the cytoplasm and makes contact with CO^2+^, and the green fluorescence weakens or disappears.

### 4.7. Calcium (Ca^2+^) Concentration Detection

Ca^2+^ concentration was detected by Fluo-4 AM kit (S1060, Beyotime Biotechnology, China) according to the manufacturer’s instructions. About 10,000 cells were obtained by an FACS Calibur system (BD Biosciences, New Jersey, BD, USA) for analysis under FlowJo software (version 10.0.7).

### 4.8. Flow Cytometry Assessment of Apoptosis

Annexin V-PE/7-AAD method was used to detect the apoptosis rate, and the specific operation process was detailed in the manufacturer’s instructions (A213-01, Vazyme Biotech, Nanjing, China). About 10,000 cells were obtained by an FACS Calibur system (BD Biosciences) for analysis under the FlowJo software (version 10.0.7).

### 4.9. Localization of the Map Effector-Protein

In order to detect the localization of the Map effector-protein, not only was the pc DNA3.1 plasmid with the *map* gene synthesized by the Sykmgene Company (Beijing, China), but His tag was also synthesized on this plasmid (named Map_His_ plasmid). MAC-T cells were seeded in 24-well cell-culture plates to grow for a 60–80% confluence. Map_His_ plasmid diluted with Lipofectamine™ 3000 (L3000150, Invitrogen, Waltham, MA, USA) was transfected into the cells. After transfection, cells were stained with anti-His (1:100, AF5060, Beyotime Biotechnology, Shanghai, China) to detect the localization of Map.

### 4.10. Animals

A total of 32 SPF-pregnant Crl:CD1 (ICR) mice aged 8–10 weeks were purchased from Charles River (Beijing, China). Moreover, these mice were given plenty of water and food, and reared in a sterile environment.

### 4.11. Mouse Mastitis Model

To verify the effect of the Map effector-protein in vivo, an *E. coli* (EPEC) infected-mouse mastitis model was established, according to the previous study [44,45]. Offspring mice were removed 4 h before the experiment. Zoletil 50 (55 mg/kg, WK001, Virbac, Carros, France) was injected intramuscularly for anesthesia, and *E. coli* (EPEC) dissolved in 30 µL of physiologic saline was injected into the nipple tube under the stereoscope. Eight mice in each group were randomly divided into four different treatment groups, as follows: (1) the control group (CONT group, an equal volume of physiological saline alone); (2) the *E. coli* (EPEC) group (WT group, WT strain 1 × 10^6^ CFU/30 μL saline); (3) the *map*-mutant group (∆*map* group, *map* mutant strain 1 × 10^6^ CFU/30 μL saline); and (4) the *map*-gene complementation-strain group (∆*map* + *pmap* group, *map*-gene complementation-strain 1 × 10^6^ CFU/30 μL saline). All mice were euthanized 24 h after infection and breast tissues were collected and stored at −80 °C.

### 4.12. Histopathology, TUNEL, and Transmission Electron Microscopy Assay (TEM)

When these mice were euthanized, the breast tissues were collected and immediately fixed in 4% paraformaldehyde/3% glutaraldehyde (pH = 7.4) for 24 h/48 h. After creating breast tissue slices, H & E staining was performed to score the pathological injury, as previously described [46]. TUNEL staining and TEM assay were performed on the breast tissue slices, according to the manufacturer’s instructions (A112, Vazyme, Nanjing, China) and the standard TEM procedure [25].

### 4.13. Data Analysis

GraphPad Prism v7 software was used to perform the statistical analysis. Data were expressed as the means ± SEM (*n* = 6 or 8). Student’s *t*-test and one-way analysis of variance (ANOVA) followed by the Tukey’s test were applied to analyze statistically significant differences at *p* < 0.05.

## Figures and Tables

**Figure 1 ijms-23-04907-f001:**
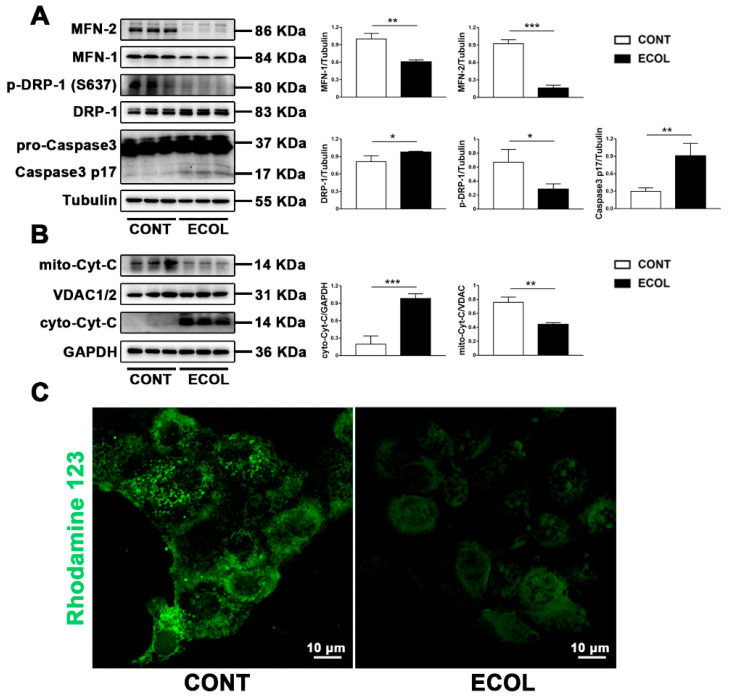
*E. coli* (EPEC) increases mitochondrial fission, decreases fusion, induces Cyt-C release and decreased MMP, and eventually leads to apoptosis. (**A**) The DRP-1, p-DRP-1 (ser637), MFN-1, and MFN-2 levels were analyzed by Western blot. (**B**) Cell mitochondria were isolated to analyze the Cyt-C level in mitochondria and cytoplasm by Western blot. (**C**) Rhodamine123 staining was used to observe the MMP changes. Scale bar, 10 µm. Representative images are shown. MAC-T cells were washed with PBS and challenged with *E. coli* (EPEC) for 8 h (MOI = 66). Data are from 3 independent experiments and presented as the mean ± SEM; * *p* < 0.05, ** *p* < 0.01, *** *p* < 0.001.

**Figure 2 ijms-23-04907-f002:**
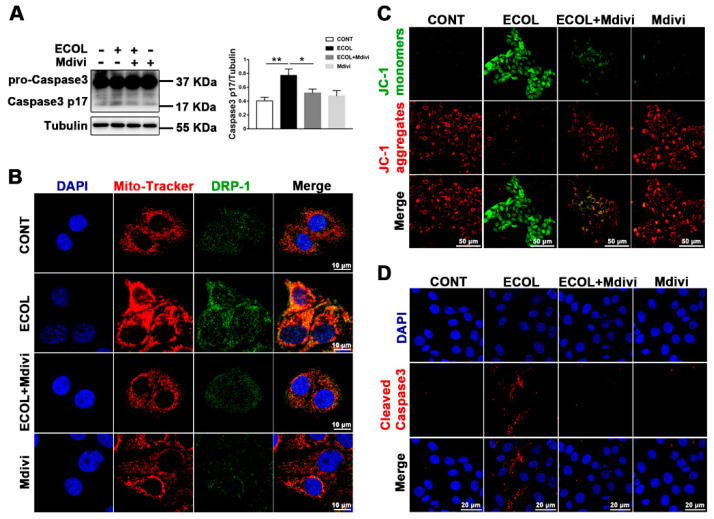
*E. coli* (EPEC) mediates DRP-1-dependent apoptosis. (**A**) The caspase-3 level was analyzed by Western blot. (**B**) DRP-1 translocated to mitochondria. Cells were stained with anti-DRP-1 (green), mitochondria (Mito-Tracker, red), and DAPI (Blue, nucleus). Scale bar, 10 µm. (**C**) JC-1 was used to analyze the MMP. JC-1 monomers, green. JC-1 aggregates, red. Scale bar, 50 µm. (**D**) Cleaved caspase-3 (red) immunofluorescence staining was used to observe apoptosis. DAPI (blue, nucleus); scale bar, 20 µm. MAC-T cells were pre-treated with Mdivi (10 μM) for 4 h, washed with PBS (3 times) to remove inhibitor, and exposed to *E. coli* (EPEC, MOI = 66) for 8 h. Representative images are shown. Data are from 3 independent experiments and presented as the mean ± SEM; * *p* < 0.05, ** *p* < 0.01.

**Figure 3 ijms-23-04907-f003:**
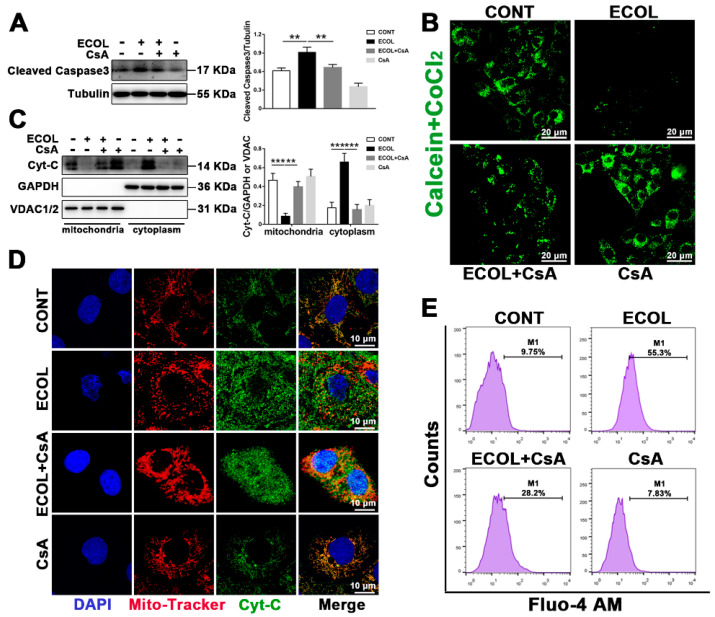
*E. coli* (EPEC) evokes the continuous opening of mPTP and mitochondrial Cyt-C and Ca^2+^ release. (**A**) The cleaved caspase-3 level was analyzed by Western blot. (**B**) The opening of mPTP was detected by the mPTP kit. Scale bar, 20 µm. (**C**) Cell mitochondria were isolated to analyze the Cyt-C level in the smitochondria and cytoplasm by Western blot. (**D**) Representative immunostaining of MAC-T cells showed increased cytoplasm Cyt-C (green) at 8 h after infection, while it was not apparent after adding CsA. Mitochondria (Mito-Tracker, red), DAPI (blue, nucleus); scale bar, 10 µm. (**E**) Quantitative Ca^2+^ concentration was detected by Fluo-4 AM flow cytometry. MAC-T cells were pre-treated with 5 μM of CsA for 2 h, washed with PBS (3 times) to remove inhibitor, and exposed to *E. coli* (EPEC, MOI = 66) for 8 h. Representative images are shown. Data are from 3 independent experiments and presented as the mean ± SEM; ** *p* < 0.01 *** *p* < 0.001.

**Figure 4 ijms-23-04907-f004:**
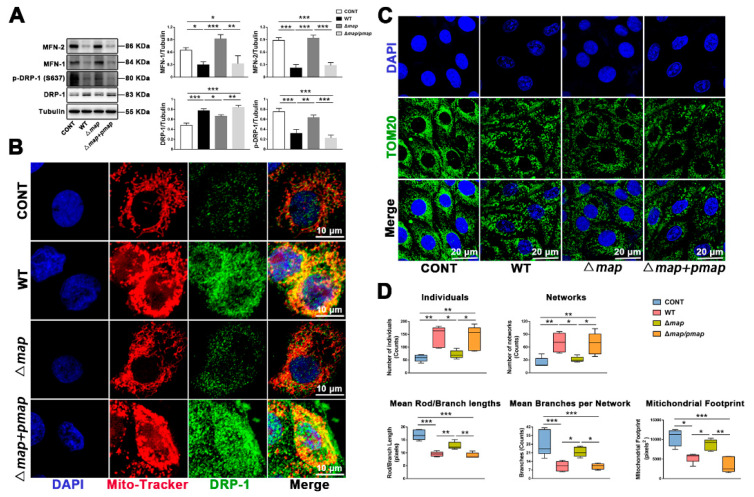
*E. coli* (EPEC)-Map induces mitochondrial fragmentation. (**A**) The DRP-1, p-DRP-1 (ser637), MFN-1, and MFN-2 levels were analyzed by Western blot. (**B**) DRP-1 translocated to the mitochondria. The cells were stained with anti-DRP-1 (green), mitochondria (Mito-Tracker, red), and DAPI (blue, nucleus). Scale bar, 10 µm. (**C**) Mitochondrial morphology was stained with anti-TOM20 (green) antibody. DAPI (blue, nucleus); scale bar, 20 µm. (**D**) FIJI software was used to analyze the related indexes of the mitochondrial network structure. MAC-T cells were washed with PBS (3 times) to remove antibiotics and exposed to WT, ∆*map*, and ∆*map* + *pmap* strains for 8 h (MOI of all strains = 66). Representative images are shown. Data are from 3 independent experiments and presented as the mean ± SEM; * *p* < 0.05, ** *p* < 0.01 *** *p* < 0.001.

**Figure 5 ijms-23-04907-f005:**
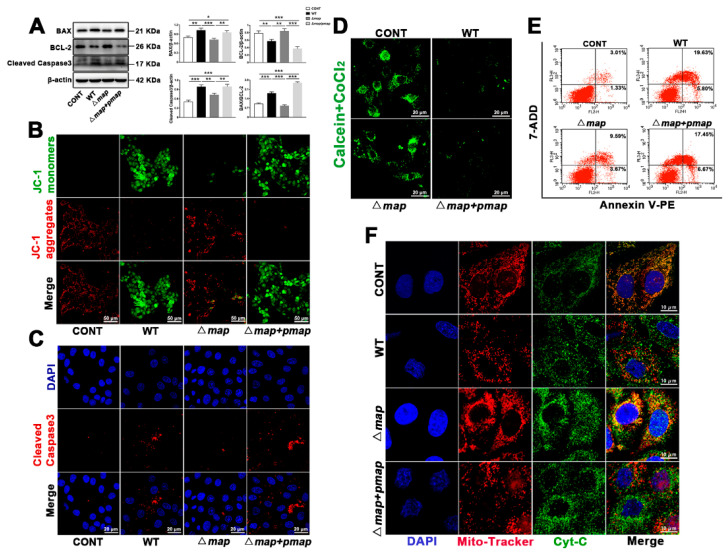
*E. coli* (EPEC)-Map results in sustained mPTP opening, Cyt-C release, low MMP, and apoptosis. (**A**) The BAX, BCL-2, and cleaved caspase-3 levels were analyzed by Western blot. (**B**) JC-1 staining was used to analyze MMP. JC-1 monomers, green. JC-1 aggregates, red. Scale bar, 50 µm. (**C**) Cleaved caspase-3 (red) immunofluorescence staining was used to observe apoptosis. DAPI was blue to indicate the nucleus. Scale bar, 20 µm. (**D**) The mPTP kit is used to detect the opening of mPTP. Scale bar, 20 µm. (**E**) The apoptosis rate was determined by Annexin V-PE/7-ADD staining assay. (**F**) Map induces the release of Cyt-C into the cytoplasm. Cyt (green), mitochondria (Mito-Tracker, red), and DAPI (blue, nucleus) are shown. Scale bar, 10 µm. MAC-T cells were washed with PBS (3 times) to remove antibiotics and exposed to WT, ∆*map*, and ∆*map* + *pmap* strains for 8 h (MOI of all strain = 66). Representative images are shown. Data are from 3 independent experiments and presented as the mean ± SEM; * *p* < 0.05, ** *p* < 0.01 *** *p* < 0.001.

**Figure 6 ijms-23-04907-f006:**
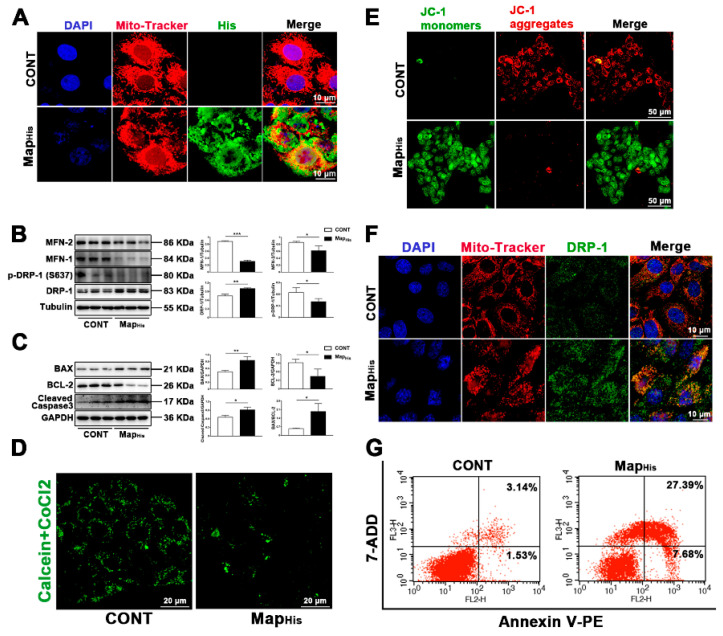
Map, as a virulence factor of *E. coli* (EPEC), is sufficient to induce mitochondrial fission and apoptosis. (**A**) Map localization to mitochondria. MAC-T cells were stained with anti-His (Map) antibody, mitochondria (Mito-Tracker, red), and DAPI (blue, nucleus). Scale bar, 10 µm. (**B**) The DRP-1, p-DRP-1 (ser637), MFN-1, and MFN-2 levels were analyzed by Western blot. (**C**) The BAX, BCL-2 and cleaved caspase-3 levels were analyzed by Western blot. (**D**) The mPTP kit was used to detect the opening of mPTP. Scale bar, 20 µm (**E**) JC-1 was used to analyze the MMP. JC-1 monomers, green. JC-1 aggregates, red. Scale bar, 50 µm. (**F**) DRP-1 translocated to the mitochondria. Cells were stained with anti-DRP-1 (green), mitochondria (Mito-Tracker, red), and DAPI (blue, nucleus). Scale bar, 10 µm. (**G**) The apoptosis rate was determined by Annexin V-PE/7-ADD staining assay. The pc DNA3.1 plasmid with *map* was synthesized by the Sykmgene Company, and the His tag was synthesized on the plasmid (Map_His_ plasmid). Map_His_ plasmid diluted with Lipofectamine™ 3000 was transfected into the cells. Representative images are shown. Data are from 3 independent experiments and presented as the mean ± SEM; * *p* < 0.05, ** *p* < 0.01 *** *p* < 0.001.

**Figure 7 ijms-23-04907-f007:**
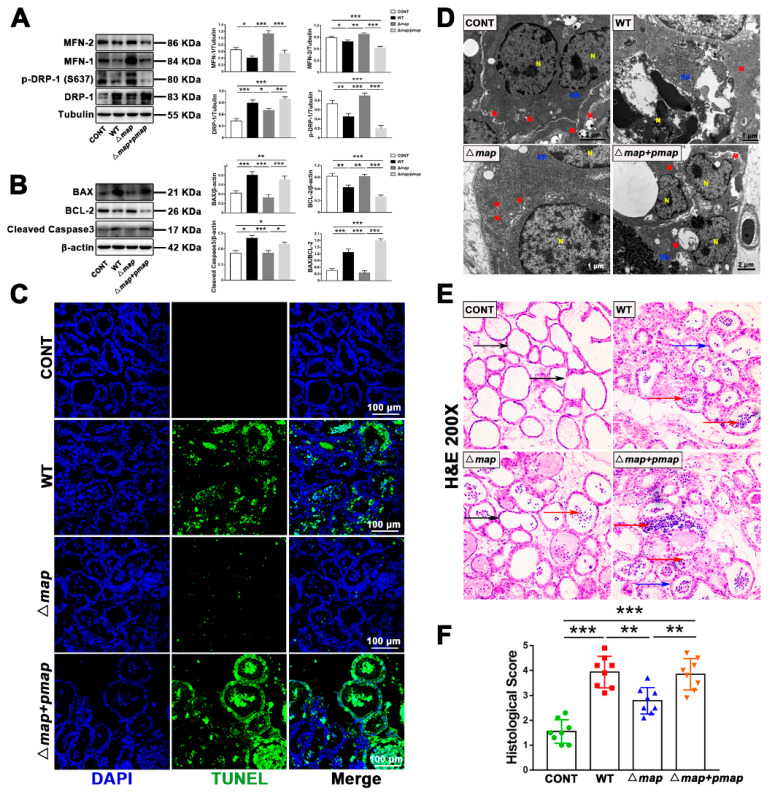
Map is required for *E. coli* (EPEC)-induced apoptosis in vivo. (**A**) The DRP-1, p-DRP-1 (ser637), MFN-1, and MFN-2 levels were analyzed by Western blot. (**B**) The BAX, BCL-2, and cleaved caspase-3 levels were analyzed by Western blot. (**C**) Apoptosis in the breast tissue was accessed through TUNEL staining and DAPI (blue, nucleus). Scale bar, 100 µm. (**D**) Representative TEM images in each group. Red M, mitochondria; yellow N, nucleus; blue ER, endoplasmic reticulum. Scale bar, 1 µm and 2 µm. (**E**) The pathological damage of breast tissue was evaluated by H & E staining (×200). The black arrow points to the normal tissues. The red arrow represents inflammatory cell infiltration. The blue arrow points to the alveolar wall hyperplasia. (**F**) Histopathological scores ranged 1–5: (1) no obvious histological changes; (2) minimal histopathological changes (i.e., individual neutrophils); (3) mild histopathological changes (i.e., a few neutrophils); (4) moderate histological injury (i.e., many neutrophils and slight glandular structure damage); and (5) severe histological injury (i.e., numerous neutrophils and severe glandular structure damage). A mouse mastitis model was established by injecting WT, *map* mutant (∆*map*), and complementary strain (∆*map* + *pmap*) (1 × 10^6^ CFU/30 mL saline) into a mammary duct 24 h after infection. These mice were euthanized and mammary tissue samples were collected. Data are from 3 independent experiments and presented as the mean ± SEM; * *p* < 0.05, ** *p* < 0.01, *** *p* < 0.001.

## Data Availability

Not applicable.

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
