# Peer review of "Map of Enteropathogenic Escherichia coli Targets Mitochondria and Triggers DRP-1-Mediated Mitochondrial Fission and Cell Apoptosis in Bovine Mastitis"

_ijms, 2022, doi:10.3390/ijms23094907_

Round 1

Reviewer 1 Report

Introduction

“So far, a total of 7 effector proteins encoded by LEE 48 virulence island have been found, namely Tir, EspB, EspF, EspG, EspH, EspZ (formerly 49 SepZ) and Map.” In a supplementary table, please described the genes that encode these proteins and please present details about the biological role of these proteins.

Please improve the final paragraph. First, please recapitulate the hypothesis of the study. Then, please clearly describe the objectives.

M & M

Please provide the strain history details (isolation, biochemical characterization, phage typing, antibiotic resistance pattern etc.)

Culture plates: did you use U or V-bottom plates? Please clarify.

I am a bit sceptical personally with the C3601 kit, as it did not provide always consistent results.

Same with caspase3 of CST

However, the authors may have had better experience than I, no problem.

The other reagents are very OK, I agree with their use.

How did you perform euthanasia of mice? Please describe.

Are you sure about normal distribution of the results? Please provide evidence. T-test is the wrong statistical test for these data, please use appropriate method.

Results

In the results section, there are many passages, which are really discussion, as they explain results. This must be corrected. Please limit the amount of information in this section, presenting only results, with no explanations. Please transfer these sentences to the discussion.

Discussion

As many of the results have already been discussed in the previous section, this part of the manuscript comes rather abruptly.

So, please move text from results to this section to allow a better flow of the text. Also, please divide the discussion into two or three sub-sections, which will allow easier reading in the final paper.

The manuscript should be reconsidered after making the corrections indicated.

Reviewer 2 Report

The current research is novel, correctly presented and it could be of very significant scientific value for scientific community and deserves to be published. The paper is well written, the text is clear. The abstract is informative and clear. Experimental design is adequate and the results are clearly presented, interesting and support the conclusions. Overall, the study provides potentially valuable data. I have no significant objections. However, I suggest that authors add several recent references in the Introduction section in order to improve it because only 10 references in total are in last 5 years.

References which I suggest in the manuscript are following:

Isolation of bovine clinical mastitis bacterial pathogens and their antimicrobial susceptibility in the Zenica region in 2017. Vet. stn. 51, 47-52. (In Croatian). doi.org/10.46419/vs.51.1.5

Bovine mastitis: a persistent and evolving problem requiring novel approaches for its control - a review. Vet. arhiv 88, 535-557. doi.org/10.24099/vet.arhiv.0116

Effect of feed additive supplementation on bovine subclinical mastitis. Vet. stn. 52, 445-460. doi.org/10.46419/vs.52.4.12

Investigation of the presence of slime production, VanA gene and antiseptic resistance genes in Staphylococci isolated from bovine mastitis in Algeria. Vet. stn. 52, 57-63. doi.org/10.46419/vs.52.1.9

Multi Locus Sequence Typing and spa Typing of Staphylococcus aureus Isolated from the Milk of Cows with Subclinical Mastitis in Croatia. Microorganisms 9, 725. doi: 10.3390/microorganisms9040725

Alternative treatment of bovine mastitis. Vet. stn. 52, 639-649. doi.org/10.46419/vs.52.6.9

Use of somatic cell count in the diagnosis of mastitis and its impacts on milk quality. Vet. stn. 52, 751-764. (In Croatian). doi.org/10.46419/vs.52.6.11

Author Response

I have added the recommended reference to the introduction.

Round 2

Reviewer 1 Report

The authors have answered well the points raised.
However, they should also add these clarifications in the revised manuscript to help future readers.
Please make changes in the manuscript before final acceptance to clarify the points raised.